# Nuclear Structure, Size Regulation, and Role in Cell Migration

**DOI:** 10.3390/cells13242130

**Published:** 2024-12-23

**Authors:** Yuhao Li, Shanghao Ge, Jiayi Liu, Deseng Sun, Yang Xi, Pan Chen

**Affiliations:** 1State Key Laboratory of Food Science and Resources, Jiangnan University, Wuxi 214122, China; zyli16895@gmail.com; 2Department of Biochemistry and Molecular Biology, School of Basic Medical Sciences, Health Science Center, Ningbo University, Ningbo 315211, Chinasundesen@nbu.edu.cn (D.S.); xiyang@nbu.edu.cn (Y.X.)

**Keywords:** nucleus, nuclear components, cellular functions, nuclear size, mechanical function

## Abstract

The nucleus serves as a pivotal regulatory and control hub in the cell, governing numerous aspects of cellular functions, including DNA replication, transcription, and RNA processing. Therefore, any deviations in nuclear morphology, structure, or organization can strongly affect cellular activities. In this review, we provide an updated perspective on the structure and function of nuclear components, focusing on the linker of nucleoskeleton and cytoskeleton complex, the nuclear envelope, the nuclear lamina, and chromatin. Additionally, nuclear size should be considered a fundamental parameter for the cellular state. Its regulation is tightly linked to environmental changes, development, and various diseases, including cancer. Hence, we also provide a concise overview of different mechanisms by which nuclear size is determined, the emerging role of the nucleus as a mechanical sensor, and the implications of altered nuclear morphology on the physiology of diseased cells.

## 1. Introduction

Eukaryotic cells differ from prokaryotic cells by having a well-defined nucleus [1]. The nucleus stores genetic information and functions as the primary regulatory hub in the cell, controlling various aspects of c ellular function. Changes in the standardized structure, morphology, or organization of the nucleus are associated with a wide range of pathological conditions. To establish and maintain proper cellular functions, most nuclear components are typically situated in their designated positions, present in precise quantities, and function at appropriate times [2]. Therefore, determining the intricate structures of these components and the specific roles they play in cellular functions is important. In this review, we discuss recent studies that have shed light on the main nuclear constituents and their respective functions, focusing on the linker of nucleoskeleton and cytoskeleton (LINC) complex, the nuclear envelope (NE), the nuclear lamina, and chromatin (Figure 1) [3,4,5,6].

Since an appropriate nuclear size is crucial for maintaining optimal cellular function, we are particularly interested in exploring the mechanisms governing nuclear size and its effect on cellular function. Here, we summarize recent advances concerning the mechanisms that regulate nuclear size, with a specific focus on regulators such as the nuclear lamina, chromatin, osmotic pressure, and others. While our primary concern is nuclear size, we also discuss the factors influencing nuclear shape, as alterations in nuclear shape may indicate alterations in nuclear size.

Interestingly, some studies have suggested that nuclei can function as mechanical sensors that can modify their morphology and regulate cell behaviors in response to environmental pressure. Herein, we examine recent advances describing how nuclei are involved in nuclear mechanics and how impairment in nuclear mechanics leads to multiple disorders. Furthermore, we briefly touch upon how alterations in nuclear morphology may potentially contribute to the development of certain diseases.

## 2. Nuclear Components and Their Cellular Functions

Any abnormal changes in nuclear structure, including genetic instability, aberrant chromosomal numbers, altered gene expression, and metabolic imbalance, can impair cellular functions [7].

### 2.1. The Structure and Cellular Functions of the LINC Complex

The LINC complex spans the inner and outer nuclear membranes of the NE, transmitting force between the cytoplasm and nucleoplasm. Its core components are SUN and KASH domain-containing proteins [8]. Specifically, proteins like nesprin-1 and nesprin-2, containing KASH domains, are anchored at the outer nuclear membrane and interact with SUN domain-containing proteins, which are embedded in the inner nuclear membrane (Figure 1) [9,10]. Several models have been proposed to understand the structure of the LINC complex. One canonical model suggests that SUN proteins can form a dimer of two trimers, i.e., a hexamer, to interact with KASH proteins [11]. Alternatively, SUN proteins can interact with chromatin independently without the involvement of KASH proteins, which are referred to as the “self-locked” [12,13]. Recently obtained crystal structures have revealed that the SUN–KASH complex adopts a 9:6 configuration in vivo [14], where three SUN homotrimers connect with six KASH proteins, suggesting the feasibility that SUN–KASH complexes can assemble asymmetrically. Also, the actin-binding motif calponin homology (CH) domain of nesprin can interact with cytoskeletal networks, thus facilitating force transduction from the cytoplasm to the nucleus and ultimately promoting nuclear movement [15]. The mechanism involves the interaction of KASH proteins with microtubules, followed by the recruitment of dynein to the outer nuclear membrane, generating the force needed to move the nucleus toward the microtubules [16]. These findings suggest that the LINC complex greatly promotes nuclear migration by connecting the cytoskeleton with the nucleus. For example, the nuclei of postembryonic cells (P cells) the vulval and neuronal precursor cells of *Caenorhabditis elegans* (*C. elegans*) migrate from lateral positions to the ventral cord through a constricted space between muscles and the cuticle during first larval development (Figure 2) [17,18]. Next, P cells differentiate to form the vulva and GABA neurons. Researchers have investigated the absence of GABA neurons to assess nuclear migration. When MET-2, a homolog of mammalian methyltransferase, which is essential for H3K9 methylation, is knocked out in *C. elegans*, heterochromatin detaches from the NE without any loss of GABA neurons. However, animals with both the LINC complex defects and MET-2 deficiency exhibit greater loss of GABA neurons than those with only the LINC complex defects. This indicates that the LINC complex may interact with perinuclear heterochromatin to regulate nuclear movement [19].

Interestingly, researchers have discovered that the loss of the H3K9-specific histone methyltransferase SUV39H1, a human homolog of the *Drosophila* protein Su(var)3-9, causes the Golgi apparatus to scatter; this process is mediated by the LINC complex [20]. The loss of H3K9me3 probably promotes the formation of the SUN2-nesprin2 complex, which connects with the plus-end kinesin family member 20A (KIF20A), thus activating its dynein activity [21]. The steady state of the Golgi complex is maintained by the minus-end-directed microtubule (MT) motor [22]. Therefore, KIF20A activation can disrupt the balance between the Golgi complex and MT dynamics, ultimately causing the Golgi complex to disperse. These findings indicate that the LINC complex facilitates bidirectional interactions between the cytoplasm and nucleus.

### 2.2. The Structure and Cellular Functions of the NE

The NE, which is composed of a double membrane with inner and outer layers, separates the nucleus from the cytoplasm (Figure 1). Nuclear pore complexes (NPCs) are large protein assemblies embedded in the NE that control transport between the nucleus and cytoplasm and ensure precise gene regulation by facilitating the exchange of necessary compounds [23,24]. Intriguingly, NPCs can also regulate the genome in response to environmental mechanical stress [25]. To gain deeper insights into the function of NPCs, a thorough investigation needs to be conducted on their structure. NPCs are composed of three layers, including the cytosolic ring, the inner ring, and the nuclear ring, and contain more than 1000 protein molecules [26]. Currently, it has been discovered that *Saccharomyces cerevisiae* (*S. cerevisiae*) has two distinctive types of NPCs in the same nucleus: one harbors a basket-like structure on the nuclear face, whereas the other does not [27]. Novel methods have been developed to isolate NPCs from the nucleus of *S. cerevisiae* for further investigation of their heterogeneous components, associated proteins, and functions. Researchers tagged GFP with nucleoporin 133 (Nup133) in basket-less NPCs and protein A with Mlp1 in basket-containing NPCs, respectively. Then, they used different beads to distinguish the tagged proteins and successively pulled down two distinct NPCs to analyze their compositions and associated proteins [28,29]. In the next step, the newly identified components of NPCs and their functions need to be determined.

Nucleoporins (Nups) are the functional subunits of NPCs and serve as their primary components [30]. The integrity of NPCs ensures the stability of nuclear circumstances, and a deficiency of NPCs can lead to cell dysfunction and even severe diseases attributed to an imbalance in cellular homeostasis. For example, focal segmental glomerulosclerosis (FSGS), which is caused by deficient podocytes, is now believed to be associated with mutations in Nup93 and Nup205 [31]. Researchers have recently reported that sufficient and proper nuclear localization of Yes-associated protein (YAP), a transcriptional activator and WW domain-containing transcription regulator 1 (WWTR1, also known as TAZ), is required for cell homeostasis [32], a process that largely depends on the accurate expression of Nup205 to facilitate the shuttling of YAP and TAZ through the NE into the nucleoplasm. Therefore, a decrease in YAP and TAZ can lead to low transcriptional activity and an increase in oxidative stress in podocytes, which may help explain the molecular mechanism underlying FSGS [33]. Additionally, a deficiency of the Nup protein Seh1 in Schwann cells impairs neuronal integrity by altering gene expression [34]. The depletion of Seh1 potentially results in attenuated H3K9me3 modification, which subsequently triggers the ZBP1-RIPK3-MLKL pathway and leads to the necroptosis of neurons. Consequently, it activates the immune response, recruits macrophages to peripheral nerves, and results in pathological inflammation.

Additionally, Nups play a crucial role in maintaining genome stability by regulating chromatin distribution. For instance, the Nup protein Elys predominantly tethers chromatin in the lamina-associated domain (LADs) ensuring the proper distribution of peripheral chromatin [35]. Once the level of Elys decreases, peripheral chromatin moves toward the interior, causing some compacted domains to unfold, which can lead to the derepression of several inactive genes in LADs. Moreover, Nup170, located in the inner ring of the NE in *S. cerevisiae*, is an important factor for tethering telomeres and repressing subtelomeric genes [36]. The possible reason could be that ablation of Nup170 leads to the upregulation of subtelomeric genes due to a reduction in proliferating cell nuclear antigen (PCNA) levels on DNA. When researchers deleted the enhanced level of genomic instability 1 (Elg1), a PCNA unloader, to increase PCNA levels on DNA, the abnormal expression of subtelomeric genes caused by the deletion of Nup170 was rescued. This occurred probably due to an increase in the level of PCNA, which can interact with chromatin assembly factor 1 (CAF-1), a protein complex that promotes the formation of heterochromatin [37]. However, the mechanisms by which Nups interact with chromatin are not fully clear. One reported mechanism involves the interaction between Nup98 and chromatin through a histone deacetylase (HDAC)-dependent pathway. After HDAC activity was inhibited, the signal of Nup98 at its chromatin binding sites decreased [38]. Nups communicate not only with chromatin but also with components surrounding the NE. NPCs cooperate with the LINC complex to form a cohesive unit, contributing to the distribution of the centromere [39]. In some Nup mutants of *Arabidopsis thaliana*, such as Nup85-1, Nup85-2, Nua-2, and Nua-3, the location of centromeres tends to rearrange, leading to their accumulation near the nuclear membrane compared to that of the wild type. These findings suggest that Nups play a crucial role in maintaining genome stability by regulating its distribution.

NPCs, which serve as pathways for transporting molecules between the nucleus and cytoplasm, contain repetitive sequences of phenylalanine–glycine (FG) that assist in the movement of proteins with a nuclear localization signal (NLS) into the nucleus or proteins with a nuclear export signal (NES) out of the nucleus. Nuclear import is considered to have two processes [40]. One process involves fast cargo transport via the exclusion zone. FG-Nups interact with each other through liquid–liquid phase separation (LLPS) to form a sieve, allowing small particles to pass through [41,42,43]. Second, the FG domains unfold, releasing the cargo into the nucleoplasm via slow cargo transport. Macromolecules containing NLS signals are recognized by karyopherin, which then facilitates protein import through NPCs into the nucleus by forming an NLS-karyopherin-FG complex [44]. Additionally, large macromolecules, such as RNA granules and aggregated proteins that exceed the exclusion limit of approximately about 40 nm of the NPCs, can be transported through NE budding, similar to the transport of herpesvirus [45,46]. In *S. cerevisiae*, Pif1, a DNA helicase, is important for maintaining nuclear and mitochondrial DNA integrity [47]. As *Δpif1* cells are difficult to culture, the *Δpif1* strain is not ideal for investigating the functions of nuclear Pif1. ^781^KKRK^784^ has been considered the core of the NLS in Pif1. Therefore, knocking out the ^781^KKRK^784^ region of Pif1 can be a feasible strategy to mimic the *Δpif1* strain [48]. This finding suggests a novel method to characterize the functions of lethal-depletion karyopherin proteins. It is intriguing that NPCs can either constrict or dilate when *Dictyostelium discoideum* cells are treated with 0.4 M sorbitol or ddH_2_O, respectively, and the diameter of NPCs is positively correlated with the flow rate across NPCs [49]. However, their permeability barriers for macromolecules in the fluid still remain intact, indicating that NPCs have adaptive mechanisms to regulate their permeabilities for macromolecules. This finding indicates that nuclei can adapt to changes in osmotic pressure while maintaining their functionality.

The NE proteins ensure nuclear homeostasis in various ways. For example, nuclear membrane proteins such as Lem2 and Bqt4 can prevent the NE from rupturing through lipid synthesis [50]. Bqt4 can recruit phosphatidic acid to the rupture site through its intrinsically disordered regions (IDRs), creating a microenvironment that facilitates the formation of the endosomal sorting complexes required for transport (ESCRT)-III complex, which helps seal ruptures and repair the NE [51]. Additionally, the nuclear cargo adaptor Atg39 mediates nucleophagy by transporting nuclear metabolites from the nucleus to the vacuole (in yeast and plants) for degradation [52], thereby maintaining nuclear architecture and thus promoting longevity [53].

To summarize, maintaining the proper state of the NE is essential for cellular processes, such as transcription, cell division, and spermiogenesis [54]. Therefore, the factors leading to NE deformation and its implications need to be identified and investigated.

### 2.3. The Structure and Cellular Functions of the Nuclear Lamina

The nuclear lamina, which is composed of lamins and lamin-associated proteins, is located beneath the inner nuclear membrane of the NE and helps maintain the stability and integrity of the nucleus (Figure 1) [55]. The aberrant arrangement of lamins is closely associated with nuclear rupture and blebbing. In vertebrates, lamins are classified into A-type and B-type. Lamin A and lamin C, encoded by the *LMNA* gene, are A-type, and lamin B1 and lamin B2, encoded by the *LMNB1* and *LMNB2* genes, respectively, are B-type [56,57,58]. Although the lamin A and lamin B meshwork form independently, progerin (a mutant form of lamin A) induces irregularities and large openings in both the lamin A and lamin B meshwork. These defects can be effectively restored by expressing lamin B1 [59]. This finding indicates that the lamin A and lamin B meshworks are somehow interconnected and interdependent.

First, the depletion, mutation, or phosphorylation of lamin A can impact the distribution of the NE proteins, nuclear morphology, and cellular function, possibly by altering the self-assembly of lamin A/C [60,61,62,63,64,65]. In mouse embryo fibroblasts (MEFs), the absence of lamin A/C can result in the mislocalization of emerin and NPCs [66]. Furthermore, the depletion of lamin A, lamin C, or lamin A/C in MEFs causes varying levels of nuclear blebbing [67]. Only the loss of lamin A increases nuclear blebbing, indicating that although lamin A and C are encoded by the same gene, they have different effects on the NE. These results indicate that lamin A is important for nuclear morphology, which is consistent with the role of A-type lamins in repairing the nuclear lamina after the rupture of NE in the mammalian nucleus [68]. The lamin A R564P mutation causes Nups to accumulate at the NE and in the cytoplasm in *Drosophila*, which leads to imbalances in protein homeostasis, slow larval motility, and muscle amyotrophy [69]. Additionally, TGFβ, which plays critical roles in epithelial–mesenchymal transition (EMT), has recently been shown to activate the Smad3 signaling pathway, which in turn activates the serine/threonine kinase AKT2, causing the phosphorylation of Ser390 at lamin A in A549 cells [70]. Inhibiting the phosphorylation of Ser390 prevents nuclear deformation, suggesting that the phosphorylation of lamin A at Ser390 contributes to nuclear modeling [71]. Second, lamin A/C plays a key role in the cellular response to mechanical strain. Mechanical stretching of muscle cells can induce the recruitment of esmin and plectin to the nuclear periphery via lamin A/C, thus maintaining nuclear morphology. However, in lamin A/C-deficient cells, the absence of interactions between lamin A/C and esmin and plectin leads to nuclear deformation [72]. Mostafazadeh et al. proposed a mathematical model to elucidate a physical modulus expressing the mechanical behavior of the nuclear lamina, in which the stretching ability of lamin A is similar to that of macromolecules with coiled-coils and extended helical motifs, such as DNA [73]. This physical modulus can assist in calculating the precise density and shear modulus of the lamin network, providing a deeper understanding of how nuclear mechanics change in laminopathies.

Lamin B participates in several biological processes, and any abnormalities in lamin B modification, whether expressed or localized, can lead to changes in chromatin organization and gene expression. Lamin Dm0 is a B-type lamin in *Drosophila*, and its distribution is impaired in the muscle cells of PIGB (phosphatidylinositol glycan of complementation class B)-deficient mutants [74]. Researchers have reported that the abnormal distribution of lamin Dm0 can increase the number of LADs, accompanied by a decrease in their average size. This alteration results in changes in gene expression, which contributes to muscle defects. Additionally, autosomal dominant leukodystrophy (ADLD) is a lethal neurological disorder characterized by demyelination in the central nervous system (CNS), and it is speculated to be caused by tandem genomic duplications of lamin B1. Recently, Nmezi et al. reported the presence of a 19 kb silencer region, specifically in oligodendrocytes, which downregulates the transcription of lamin B1 [75]. The silencer region maintains the homeostasis of lamin B1, protecting individuals with tandem duplications of lamin B1 from developing ADLD. Their finding suggests that additional silencer factors may be absent or dysregulated in ADLD patients. To summarize, both the deficiency and overexpression of B-type lamins can have severe consequences on neural development. Additionally, B-type lamins play critical roles in maintaining chromatin stability, thus preventing abnormal transcription and ensuring proper cellular function [76].

Although lamins are specific to metazoan cells and have evolved to adapt to diverse environments, numerous lamin-like proteins have been identified in non-metazoan species [77,78]. These proteins exhibit many similarities to lamins in terms of structural organization, conserved regions, and subnuclear distribution, and fill the functions of lamins, such as chromatin regulation, in plants [79] or lower eukaryotes [80]. Thus, researchers sought to investigate the functions of these related proteins to better understand their roles across different species [66]. They selected the *Dictyostelium* lamin-like protein NE81 and expressed either NE81 or lamin A in *Lmna^–/–^* MEFs and triple lamins knockout (TKO) MEFs, which lack all lamins, to investigate the function of NE81. They reported that the expression of NE81 in *Lmna^–/–^* MEFs and TKO MEFs partially rescues nuclear deformability, but it is not as effective as lamin A in restoring normal function. These findings indicate that differences in self-assembly or self-interaction exist between non-metazoan and metazoan lamins. The lamin-like proteins CROWDED NUCLEI 1 (CRWN1) and CRWN2 in *Arabidopsis* interact with the DNA damage repair proteins RAD51D and SNI (suppressor of npr1-1, inducible 1) and promote the DNA repair response to DNA damage [81]. It suggested that lamins may facilitate the dynamic process of chromatin undergoing LLPS. These results indicate that lamin-like proteins exhibit similar functions, although not identical to those of canonical lamins.

Nuclear constituents are not isolated entities; rather, they are extensively interconnected as part of an integrated network. Chromatin can interact with the lamina via LADs [82]. LADs are almost inactive and rich in repressive histone modifications, such as H3K9me2/3 and H3K27me3 [83]. However, the precise role of the cross-talk between chromatin and the nuclear lamina within LADsremains incompletely understood. Pujadas Liwag et al. investigated this issue and reported that lamin B is involved in the suppression of DNA transcription [84]. Using auxin-inducible degron technology to ablate lamin B1 and lamin B2, researchers have observed the movement of peripheral chromatin into the interior, as detected by partial wave spectroscopic (PWS) microscopy. They also identified distinct alterations in gene expression in both LADs and non-LADs. Specifically, the differentially expressed genes (DEGs) in LADs are found to be associated primarily with laminopathies and chromatin structural disorders, such as scoliosis, whereas the DEGs in non-LADs are linked to malignancies. These results indicate that B-type lamins are essential for maintaining the three-dimensional genomic architecture within LADs.

Additionally, chromatin itself is required for preserving the structure of LADs. Researchers deleted specific regions ranging from 150 kb to 1.3 Mb within the LAD region on chromosome 9 in mouse embryonic stem cells (mESCs) [85]. They observed that the deletion region of LADs can autonomously associate with the nuclear lamina to different degrees. Interestingly, those remaining LADs, after rearrangements, can connect with nearby intervening inter-LAD (iLAD) and subsequently facilitate the establishment of new connections between the iLAD and the nuclear lamina, indicating that LADs can drag neighboring iLAD sequences toward the nuclear lamina. Additionally, certain iLAD that can interact with the nuclear lamina present elevated levels of H3K9me3, whereas others do not. This finding indicates that H3K9me3 deposition may serve as a marker of LAD–nuclear lamina interactions, although other mechanisms can also affect LAD–nuclear lamina interactions.

In conclusion, lamin is required for numerous cellular functions, particularly in maintaining chromatin structure. Alterations in lamin modification or expression can result in abnormal gene expression, leading to cellular dysregulation and severe diseases.

### 2.4. The Structure and Cellular Functions of Chromatin

Besides traditional membrane-bound organelles, several membrane-less organelles composed of biomolecules can undergo phase separation. Moreover, proteins containing IDRs show a similar phenomenon known as LLPS [86,87,88]. Although the dynamics of macromolecules are not yet fully understood, *Xenopus* extract offers a robust system for studying the biochemistry and biophysics of living systems [89]. Interestingly, chromatin exhibits notable Brownian diffusion during interphase, albeit its movement is restricted to specific subregions referred to as chromosome territories [90,91]. Specifically, on a larger scale, there are two types of territories: euchromatin with low DNA density and heterochromatin with high DNA density [92]. The chromatin region can also be sorted into A (active) and B (inactive) compartments [93], which almost correspond to euchromatin and heterochromatin, respectively (Figure 1) [94]. These regions occur probably due to interactions between chromatin and the nuclear lamina, as well as interactions within the chromatin [95]. Various protein and RNA factors in the nucleus can interact with each other via LLPS to promote the formation of B compartments [96]. For instance, H3K9me3 is an epigenetic marker of heterochromatin that can bind to the chromodomain of heterochromatin protein 1 (HP1) and SUV39H1, serving as a binding site for HP1 [97]. When the HP1-SUV39H1 complex connects with H3K9me3 and undergoes LLPS, it can induce the B compartments mentioned above [98]. Chromatin states can also be altered by histone posttranslational modifications (PTMs) [99]. For example, researchers substituted H3.3 K9 for A in mESCs and generated a new line known as K9A. They reported a significant reduction in H3K9me3 signals within heterochromatin and an increase in overlapping H3K27ac signals, which is a marker for active enhancers, within the same region in K9A mESCs [100,101]. Consequently, cryptic cis-regulatory elements at endogenous retrovirus sequences in heterochromatin become active, inducing the expression of immune-related genes. In sum, the chromatin state is closely linked to the level of gene expression.

Recently, researchers have started developing new techniques to investigate chromatin dynamics, which exhibit features reminiscent of LLPS. For example, some researchers have applied the VECTOR (ViscoElastic Chromatin Tethering and Organization) system to examine the movement of telomere loci, providing a novel platform for viewing chromatin dynamics [86]. To be more specific, researchers can reposition telomere loci by fusing both synthetic light-controlled condensates and telomere loci with the same phase separation-prone IDRs. Thus, the interaction between IDRs can connect two elements through capillary forces, allowing nucleic acids at telomere loci to follow the movement of light-controlled condensates [102]. Because the sequences and charges in synthetic condensates differ from those of telomere loci, they prevent the mixing of these two elements. Using this system, researchers can calculate the dynamic parameters of chromatin and find that the viscoelastic resistance of the internal locus is lower than that of its peripheral partner locus, indicating that chromatin possesses dynamic and stable properties [103]. Additionally, different chromatin densities result in distinct nonequilibrium processes, with lower density leading to a sol-like network and higher density leading to gel-like networks [104]. It is intriguing that transcription factors can exert capillary forces by condensing on DNA, which helps remodel chromatin and regulate gene expression [105,106,107].

Since chromatin contains most of the DNA present in the cell, the proper dynamics and state of chromatin are essential for maintaining nuclear homeostasis [108]. First, chromatin dynamics affects many aspects of nuclear function. For instance, during mitosis, chromatin undergoes intense compaction to prevent DNA replication and transcription [109]. Interestingly, a study reported that the transcription factor Ace2 can bind to the promoter region of the eng1 gene to decompact local chromatin during mitosis, allowing DNA activation in fission yeast [110]. It is possible that chromatin assembly through chromatin dynamics limits the accessibility of DNA and thus hinders nuclear events such as transcription, DNA replication, and DNA repair, consequently affecting cellular function. For example, a recent study revealed that nuclear actin can regulate the rearrangement of heterochromatin by altering chromatin accessibility [111]. The multipotent bone marrow mesenchymal stem cells (MSCs) were exposed to CK666 or cytochalasin D (CytoD), both of which inhibit actin polymerization, with the latter being unable to degrade nuclear F-actin. Treatment with CK666 decreases the level of nuclear F-actin, leading to increased chromatin accessibility and adipogenesis of the cells. Conversely, treatment with CytoD increases the level of branched nuclear F-actin but decreases chromatin accessibility and osteoblastogenesis. Moreover, H3K27ac levels in chromatin regions increase upon β-actin depletion [112]. A possible mechanism involves lncRNA maternally expressed gene 3 (Meg3), which can increase chromatin transcription through interactions with chromatin-modifying enzymes or transcription factors [113]. It has been reported that the level of H3K27ac in the Meg3 promoter is upregulated upon β-actin depletion, which can increase the accessibility of the Meg3 promoter and its expression. Subsequently, the expressed Meg3 moves and accumulates at sites harboring high levels of H3K27ac to regulate chromatin organization. Remarkably, computer simulations of chromatin dynamics have highlighted the crucial role of nucleosome spacing in chromatin organization and accessibility [114]. Second, the state of chromatin regulates various cellular processes, particularly transcription. For instance, treatment of pancreatic β-cells with bone morphogenetic protein-2 (BMP-2) decreases insulin release in response to glucose, attributed to PTMs [115]. This effect is caused by BMP-2-induced reductions in H3K27ac levels, leading to the formation of heterochromatin and downregulation of NeuroD1 binding sites. The decrease in the number of binding sites of NeuroD1, a transcription factor that interacts with the promoters of crucial genes for pancreatic β-cell determination, results in a decline in the number of pancreatic β-cells and, consequently, insulin secretion. Another study revealed that the precise concentration and location of H3K27ac ensure the normal process of early human embryogenesis. During early human embryogenesis, a large portion of the genome is enriched with H3K27ac, which is predominantly localized in promoters and partially methylated domains before the eight-cell stage due to the dynamics of H3K27ac [116]. Immediately after zygotic genome activation at the eight-cell stage, most of the H3K27ac is removed by histone deacetylases. This dynamic modulation of H3K27ac within chromatin causes alterations in gene expression, indicating that temporal gene regulation is important for early embryonic development. Interestingly, the length of the cell cycle can influence both the removal and reaccumulation of H3K9me3 during early development in medaka, zebrafish, and *Xenopus* embryos [117]. It is reported that a prolonged cell cycle permits the accumulation of Setdb1, subsequently initiating the onset of H3K9me3 deposition for heterochromatin formation. These results indicate that alterations in chromatin dynamics or state can regulate DNA expression, ultimately influencing cellular functions.

## 3. Mechanisms of Nuclear Size Regulation

The size of the nucleus strongly influences various processes, including development, migration, and disease progression [118,119,120]. Hence, numerous studies have attempted to elucidate the regulatory mechanisms governing nuclear size. Many factors, including nuclear transport [121,122,123], NPCs [124,125,126], the endoplasmic reticulum (ER) [127], the cytoskeleton [128], limiting components [129], the nuclear lamina [130,131,132], and chromatin [133], serve as key factors. In this section, we review some recent advancements in understanding the mechanisms by which these regulatory factors control nuclear size.

For closed mitosis, where the nuclear envelope remains intact before chromosome segregation, the size of the nucleus increases to accommodate the duplicated DNA as a natural part of cell growth and mitotic division [134]. Consequently, the amount of DNA is an important regulator of nuclear size. In different species, a strong correlation is found between DNA content and nuclear size [135,136]. In *Xenopus*, an increase in the amount of DNA contributes to nuclear size enlargement in a cell-free system [137]. This finding is also consistent with aneuploid cancer cells. Multiple copies of chromosomes typically in cancer cells have larger nuclei than those in healthy diploid cells [138]. Moreover, nuclear size is correlated with a severe state of cancer [139]. In tetraploid cells commonly found in human tumors, the G1 phase is too short for adequate formation of replication factors [140]. This can increase the number of γH2AX foci in chromatin, which is an early marker of DNA damage, making the cells unstable and prone to DNA damage, particularly in the S phase. Moreover, the double amount of DNA is densely packed within the limited nuclear space, which restricts access to DNA repair molecules, further exacerbating damage. Additionally, alterations in chromatin organization are closely related to changes in nuclear shape and disease. The degree of chromatin compaction can alter nuclear morphology [141,142]. Viola et al. reported that the compactness of chromatin influences membrane invagination during mitosis [143]. They treated cells with Calyculin A to induce chromatin condensation and reported that the cells presented a decrease in nuclear radius and an increase in the number of NE invagination events. Rather, the decompaction of chromatin can lead to the formation of nuclear blebs [144]. Similarly, in Trichostatin A-treated cells, heterochromatin loosens, promoting nuclear bleb formation [145]. Interestingly, chromatin compaction contributes to increased nuclear size in melanoma [120]. Therefore, both the amount and structure of chromatin play important roles in regulating nuclear size and are related to cancer. A mechanism by which chromatin regulates nuclear size involves alterations in chromatin–lamina interactions. In the *Xenopus* cell-free system, nuclear expansion preferentially occurs at sites of high-density chromatin and lamin addition, suggesting that peripheral chromatin–lamin incorporation generates forces pushing against the NE, which may promote nuclear growth [123].

Interestingly, in a yeast model, a 16-fold change in nuclear DNA content does not significantly change the nuclear size [146]. Moreover, during embryogenesis in most species, the nuclear size of blastomeres decreases with concomitant reductions in cell size, whereas blastomeres contain the same amount of DNA [118,147]. Finally, in two species of *Xenopus*, cytoplasmic factors exerted a more significant influence on nuclear size than DNA content [121]. These studies suggest that there are other regulators of nuclear size, like lamins and cytoplasmic volume. First, the phosphorylation of lamins is crucial in regulating nuclear size and is required for NE rupture during mitosis [148]. Specifically, researchers found that the lamin A R249Q mutation in human induced pluripotent stem cell-derived cardiomyocytes (iPSC-CMs) leads to a significant reduction in lamin A/C levels at the NE compared to those in healthy controls, resulting in an increased nuclear cross-sectional area and disruption of the NE [149]. Interestingly, the interaction of lamin A and histone H3 is considered to be a determinant of nuclear size and shape. For instance, lamin A monomers can bind directly to histone H3, which is influenced by histone PTMs [150]. Specifically, lamin A is more likely to bind to H3 with a methyl-methyl modification such as H3R8me2/K9me2 than to H3 with only a single methyl modification such as H3K9me2. Additionally, when H3 undergoes oncogenic mutations in histone H3.3, which impairs H3K27 methylation, it results in a decrease in both the circularity and size of the nucleus and can even cause severe diseases such as pediatric glioma and chondrosarcomas [151]. Second, the limiting component model explains that cytoplasmic volume contributes to nuclear size regulation [152]. If more components are present in the cell that can facilitate nuclear builds, it leads to a larger nuclear size. The nucleoplasm (Npm2) has been found to be a limiting component, where a greater amount of Npm2 correlates with a larger nuclear size [129]. Similarly, more active nucleocytoplasmic transport of components strongly influences the nuclear and cell growth rates [153]. Consequently, the nucleolus is considerably enlarged in cancer cells to meet the high demand for ribosomes due to the high proliferation rate and cell growth, which require an increase in the synthesis of new proteins [154,155]. In clinical practice, hypertrophy of the nucleolus serves as a histopathological diagnostic marker for cancer transformation and tumor progression [156].

However, a comprehensive mechanism for regulating nuclear size has yet to be fully elucidated. From our perspective, as with others, after nuclear assembly during mitosis, nuclear size is maintained through a balance of forces exerted on the outer and inner nuclear membranes; stronger pushing forces compared to pressing forces induce nuclear expansion and vice versa (Figure 1) [157]. For example, there is a tug-of-war relationship between the membranes of the NE and ER, and the ER is continuous with the outer membrane of the NE [158]. An increase in the perinuclear ER contributes to the growth of the nuclear surface, potentially by enhancing the pulling force on the NE. Additionally, an increase in nuclear influx or inhibition of nuclear efflux is accompanied by nuclear volume expansion, potentially through the augmentation of the intranuclear pushing force exerted by osmotic pressure [159]. Similarly, the increase in nuclear osmotic pressure induced by proteins and mRNAs is responsible for driving nuclear expansion [157]. Conversely, hyperosmotic pressure drives chromatin compaction, resulting in small and stiff nuclei in *Arabidopsis* root meristems [160]. It is possible that the force pressing on nuclei from hyperosmotic pressure is stronger than the intranuclear pushing force generated from compact chromatin, resulting in smaller nuclei. One mechanosensitive sensor discovered is the Piezo channel, which can detect fluid stress in the environment [161]. However, further evidence is required to validate this model, such as the sensor used by nuclei to detect intranuclear forces and methods to detect those forces on nuclei and other related factors.

## 4. The Nuclear Mechanical Function

The nucleus is a key sensor of the local microenvironment. When a cell is confined or subjected to mechanical stress, the increase in NE tension triggers a signaling cascade for actomyosin contractility, allowing the cell to migrate out of confinement [162,163]. Additionally, the nucleus can act as a piston, exerting pressure on the front of the cell [164], thus influencing the efficiency of cell migration [165]. Deformation of the nucleus under mechanical stress can reorganize the chromatin state and increase the condensation of proteins in the nucleoplasm, thus influencing cellular function [166]. Conversely, the decompaction of chromatin can enhance the strength and contractility of cellular adhesion via RhoA activation, showing a reverse mechanotransduction pathway from the nucleus to the cell surface [167]. It is argued that the mutation E145K of lamin A leads to distinct mechanical properties of the lamina, which in turn affects Hutchinson-Guildford progeria in humans [168]. These findings highlight a clear link between the mechanics of the nucleus and the development of disease.

When cells are subjected to compression from surrounding tissues, physical force is transmitted from the extracellular environment through the extracellular matrix to the cytoskeleton and then to the nucleus, potentially deforming the NE [169]. For instance, reduced Arp2/3 activity, low adhesiveness or rigidity of the substrate, and high contractility promote bleb formation, whereas the expansion of blebs is driven by hydrostatic pressure regulated by contractility [170]. Moreover, *C. elegans* filamin FLN-2 functions along with the LINC complex and CDC-42/actin-dependent pathways to preserve the integrity of the NE and facilitate the movement of P-cell nuclei through narrow spaces (Figure 2) [171]. The deficiencies of FLN-2 increase the possibility of NE rupture and lead to the failure of nuclear migration. Therefore, nuclear morphology is generally modified by mechanical stimuli during cell migration. One potential mechanism by which mechanical forces affect cellular function is through the modification of chromatin states within the nucleus. A recent study delivered different mechanical strains on MSCs cultured on a flexible membrane via sine or brachial waveforms at different frequencies (0.1 Hz and 1 Hz). MSCs under moderate strain amplitudes (7.5–12.5%) are suitable for chromosome unwinding and expressing regenerative properties; strain amplitudes below 7.5% cannot unwind the chromosome sufficiently, whereas high amplitudes may damage the chromosome [172]. Interestingly, cellular pressure can affect the position of the nucleus. During *C. elegans* gonadogenesis, the position of the nucleus is crucial for maintaining the integrity of leader cells [173]. Overall, both the morphology and position of the nucleus are important factors for the mechanical role of the nucleus.

## 5. Nuclear Morphology and Disease

As the nucleus is the largest and firmest organelle, nuclear deformation may be related to cell migration through narrow channels, including embryonic development, wound healing, immune responses, neurodegenerative disorders, and cancer metastasis [120,174,175]. Furthermore, the association between actin and the nucleus is essential for efficient cell invasion because it provides the necessary forces for cell movement [176].

Cancer cells with high levels of nuclear deformability can gain access more easily to other tissues in the body. A study revealed that WD repeat-containing protein 5 (WDR5), a core subunit involved in H3K4 methylation, could affect the metastasis of acute lymphoblastic leukemia (ALL) cells in dense 3D conditions [177]. After inhibiting the expression of WDR5, nuclear deformability is abrogated, and the infiltration of ALL cells into other tissues (in vivo experiments) is decreased, indicating the role of WDR5 in controlling ALL cell metastasis through changes in nuclear morphology. A study on colorectal cancer cells revealed that high expression of ErbB4 leads to activation of the ErbB4-Akt1-Lamin A/C pathway, resulting in the phosphorylation of lamin A/C [178]. This, in turn, promotes the disassembly of nuclear lamina and increases nuclear deformability, ultimately leading to metastasis. Intriguingly, another investigation reported that tumor cells with high levels of lamin A are more likely to metastasize to the lymph nodes, whereas tumor cells with low levels of lamin A are more prone to enter apoptosis, possibly because low levels of lamin A lead to a fragile NE [179]. Emery–Dreifuss muscular dystrophy (EDMD) is collectively known as a disorder of nuclear envelopathies and laminopathies. One of its subtypes is related to the downregulation of the NE protein Net39, which subsequently affects NE integrity and induces DNA damage [94]. Thus, gaining a deeper understanding of NE deformation may facilitate the identification of specific targets for treating disorders related to NE morphology.

To gain more intuitive insights and understand the molecular details of nuclear morphology and mechanobiology, 3D models have been developed [180,181]. Compared to traditional 2D projections, 3D models overcome the limitations of losing important parameters, such as curvature, shape index, and fractal dimension, thus providing further mechanistic insights into cell phenotypes during migration. For example, in a 3D culture device, the use of agarose as a base for sustaining long-term cell confinement can help scientists investigate cell behaviors throughout the entire cell cycle in a simulated in vivo environment [182]. In a mouse model, research revealed that prolonged exposure to adverse external stimuli can lead to NE deformation, which in turn causes cells to deteriorate [183]. These findings suggest that nuclear deformation can arise not only from intrinsic cellular factors but also from severe external environmental conditions. To summarize, understanding the normal traversal of nuclei through constricted spaces can help improve our understanding of metastasis.

Based on the aforementioned model that nuclear size is regulated through force equilibration, we speculate that additional forces may be exerted on the NE by nuclear constituents, the surrounding cytoplasm, or other factors, disrupting equilibrium and consequently resulting in the deformation of the NE. For example, increased chromatin compaction may result in greater intranuclear force, ultimately leading to the deformation of the NE [123]. However, the detailed mechanism underlying the deformation of the NE and its implications for related diseases need to be elucidated.

## 6. Conclusions

The proper nuclear architecture, including NE-associated proteins, nuclear transport, nuclear-cytoskeletal contacts, the nuclear lamina meshwork, and chromatin organization, is critical for cell function. In this review, we discuss the effect of certain nuclear components on nuclear morphology and cell function, the potential mechanisms underlying nuclear size regulation, and the mechanical role of the nucleus in cell migration. While the underlying mechanisms that regulate nuclear morphology have been elucidated in different systems, substantial gaps in our knowledge remain. Advanced microscopy technology, proteomics and genomics, single-cell technology, and other emerging technologies can help us better understand the effects of nuclear functional architecture on nuclear mechanics, chromatin organization, gene expression, and cellular functionality. Moreover, expanding the study to in vivo models to investigate nuclear mechanics during cell migration may provide valuable insights. As the size and shape of the nucleus can significantly impact various human disorders, such as cancer, aging, and different types of neuromuscular diseases, whether nuclear morphology can serve as a novel therapeutic target, particularly in cancer treatment, needs to be investigated.

## Figures and Tables

**Figure 1 cells-13-02130-f001:**
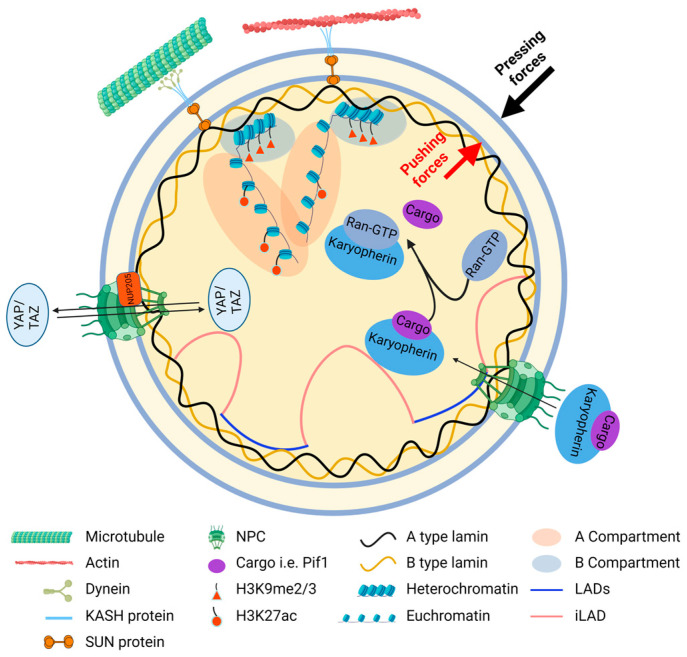
The primary structure of the nucleus. The nucleus is enveloped by the NE, upon which pushing and pressing forces are exerted. NPCs are inserted into the NE, through which karyopherin proteins can transport cargo into the nuclei. The nuclear lamina is a network beneath the NE, mainly composed of type A and type B lamins. Chromatins can be classified into A and B compartments, corresponding, respectively, to euchromatin and heterochromatin. The LINC complex is formed by proteins containing KASH and SUN domains, which are located at the outer and inner nuclear membrane, respectively. The LINC complex connects the cytoskeleton (microtubules and actin) and the nucleus. The LADs are chromatin regions that are associated with the nuclear lamina, while iLAD refers to the intervening inter-LAD. Created in https://BioRender.com.

**Figure 2 cells-13-02130-f002:**
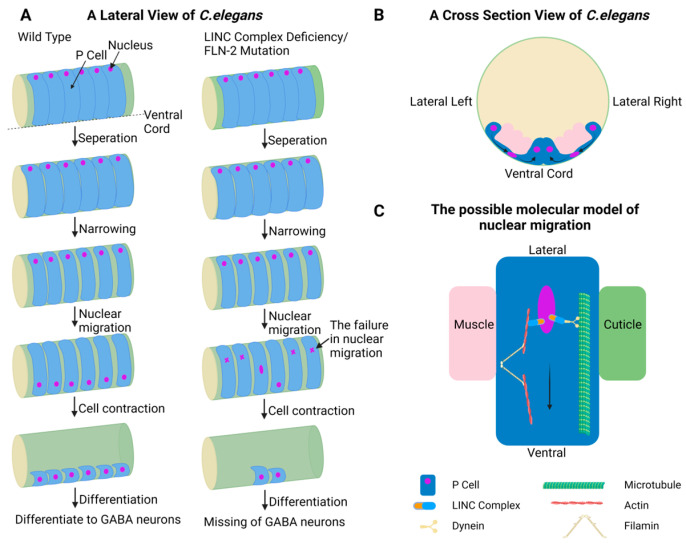
Nuclear migration in P cells of C. elegans. (**A**) From a lateral view of C. elegans, first, P-cells separate from each other; second, P cells narrow; third, the nuclei of P cells migrate from the lateral to the ventral cord; fourth, the cytoplasm of P cell retracts to the ventral cord; last, P cells develop into GABA neurons. LINC complex deficiency or FLN-2 mutant C results in the failure of nuclear migration and P cell differentiation. (**B**) From a cross-section view of C. elegans, the nuclear shape is altered through a constricted space between muscles and cuticle during migration. (**C**) One possible molecular model of nuclear migration. The nucleus migrates from the lateral to the ventral cord, relying on the deformability of the NE and the interaction between the LINC complex and the cytoskeleton. Created in https://BioRender.com.

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
