# Peer review of "Nuclear Structure, Size Regulation, and Role in Cell Migration"

_cells, 2024, doi:10.3390/cells13242130_

Round 1
Reviewer 1 Report
Comments and Suggestions for Authors
Li and co-authors provide yet another review article on nuclear architecture. There are already many current articles on this topic, so another review should stand out a little from the existing articles. This is only partially the case here. The title is very general and suggests a comprehensive review of almost all aspects of the cell nucleus, which naturally cannot be the case in around 10 pages of text. I would suggest choosing a better title that describes more clearly what this review is about. The article is tedious to read, as it completely lacks figures and is sometimes difficult to read and understand due to numerous grammatical errors. In a revised version, the use of articles, tenses and singular/plural should definitely be corrected by someone with a very good knowledge of English. I cannot list the countless errors here individually, as this would go beyond the scope of my review. In general, it should be checked throughout the text whether all abbreviations are explained (this does not seem to be the case, e.g. IDR, CH...). Furthermore, in some places in the text, instead of citing review articles by other authors, it would be better to cite the original papers on the experiments and conclusions described in the text. I highlight this only in some instances in the following specific points of critique.
- Line 74 and following: Cite original literature; the “self-locked” structure is a dimer of two trimers, i.e. a hexamer. However, there are other oligomeric structures.
- The authors refer to the C. elegans model several times in the text. For readers who are not familiar with this model, an illustration would be helpful.
- Line 94: “of” instead of “that”
- Line 111: “compounds” instead of “substance”
- Line 115: 1000 subunits is misleading. There are only about 36 subunits, most of which occur in multiples of eight.
- Line 136: “neuronal” instead of “neurons”
- Line 173: Elsewhere, the authors describe LLPS phenomena. It would therefore also make sense to point out here that the transport through the FG hydrogel is based on an LLPS process.
- Line 184: This passage is misleading. “That” instead of ‘there’. It is not clear in which direction the changes go. Which fluid is meant? I assume it is about cytoplasm vs. nucleoplasm. In principle, “in the fluid” can simply be omitted.
- Line 197: this statement is too trivial and can be omitted.
- Line 199: here it is unclear what the authors want to express. There are certainly enough researchers and more are not necessarily urgently needed. I assume it's about research, not researchers.
- Line 202: I think it would make more sense not to insert new abbreviations for the lamin proteins. Lam is unusual, why not the introduced Lmn?
- Line 206: Names of mammalian cells should be written in italics and capitalized throughout
- Line 213: There are countless papers on the topic of phosphorylation of lamins. Citing only one original paper is not sufficient.
- Line 252 to the end of the paragraph: Here it should first be clearly defined what lamin-like and what canonical lamin means. A distinction must be made between proteins that are clearly evolutionarily related to lamins, such as Dictyostelium NE81, and those that only fulfill a lamin-like function, such as the CRWN proteins of higher plants. NE81 is a lamin according to everything that is known and should now also be referred to as such or be counted among the canonical lamins.
- Line 265: Omissions by apostrophes should generally be avoided.
- Line 269: It would be better to mention the names of the groups instead of just talking about researchers.
- - Line 296: replace “varieties of” with “several”
- Line 304: I would speak of A-type and B-type and not only of A and B.
- Line 313: better: “...induce B-type chromatin compartmentalization as mentioned...”
- Line 330: “What's more” can be omitted
- Line 337: the sentence is difficult to understand. Better: “TFs can exert capillary forces by condensing on DNA, which helps to remodel chromatin and regulates gene expression”. Cite original literature!
- Line 351: Explain toxins and their effects.
- Line 359: “upon” instead of “on”
- Line 394: NE ruptures always occur in mitosis, after all, there is a NEBD in mammalian cells! It must be made clear here what is meant by this.
- Line 428: it should read: “adequate formation of replication factors”
- Line 435: “alteration” instead of “alternation”
- Line 444: “outer” instead of “out”
- Line 453: sensor
- Line 460: The nuclear envelope is normally not folded. What does unfolding mean here?
- Line 468: unclear: the mutation is NOT due to different...
- Line 479: make it clear that the sentence refers to the C. elegans model.
- Line 484: not the study found out, the research team has found out
- Line 498: explain WDR5
- Line 522: replace "research" by "study".
- Line 530: "other factors" instead of "others"
- Line 544: omit "the impact of"
- Line 546: "nuclear mechanics" instead of nuclear mechanistic function
- Line 548: omit "accelerated"

language problems are discussed in the "Comments and Suggestions for Authors
"
Reviewer 2 Report
Comments and Suggestions for Authors
In the presented manuscript, Li and colleagues discuss updated views on the role of nuclear lamina on the structural organization of the cell nucleus. Furthermore, they explore chromatin organization and structure under influence of liquid-liquid phase separation. Importantly, they pay a special attention to the mechanistic principles of regulation of the size of the nucleus.
This is a long review of an interesting topic. The manuscript needs some additional work before it is accepted for publication.
Specific points are highlighted below:
The manuscript will greatly benefit from the schematics on major topics of the review.
Line 202: add: composed of lamins (Lam) and lamin-associated proteins
Line 310: add: H3K9me3 is a epigenetic marker of heterochromatin
Line 312: add: SUV39H1 acts as a "writer/methyltransferase" by adding a methylation mark to lysine on histone H3, which then serves as a binding site for HP1.
Line 386 and below: The regulation of the nuclear size is a critical complex issue for our understanding of cell function. Therefore, this part of the manuscript should be expanded and more discussed. Here are some key points about nucleus size increase, which should be included into the manuscript.
A nucleus size naturally increases primarily during the S phase of the cell cycle, when DNA replication occurs, causing the nucleus to expand to accommodate the duplicated DNA as a natural portion of cell growth and mitotic division (doi: 10.1038/nrm3629).
DNA content influences the volume of the nucleus, which in turn influences the size of the cell. Instinctively, DNA content can affect nuclear volume, because the size of the nucleus could be directly proportional to amount of DNA it contains and the extent to which that DNA is compacted. Simply comparing genome size the nuclear volume among species supports this assumption, because species with larger genomes generally have larger nuclear volumes (doi: 10.1093/aob/mci010).
This phenomenon is also supported by observation that aneuploid cancer cells with multiple copies of chromosomes typically have larger nuclei in contrast to healthy diploid cells due to the increased amount of genomic DNA in the nucleus, which also requires a larger nuclear membrane (doi.org/10.3389/fcell.2022.1017588). Thus, when a cell grows (or getting more DNA through aberrant mitotic division in cancer) and its nucleus enlarges, the nuclear membrane expands to accommodate the increased nuclear volume.
Furthermore, it has been documented that the cell volume is tightly correlated with the nuclear volume, and active nucleocytoplasmic transport of components strongly influence the cell growth rate (DOI: 10.1371/journal.pcbi.1009400). Fundamental role in this phenomenon plays ribosome biogenesis tightly coordinated with protein synthesis which influences size of cell and nucleus. Production of ribosomes is orchestrated in the nucleolus, a membraneless organelle inside the nucleus, which is significantly enlarged in cancer cells to satisfy the high demand for ribosomes due to high proliferation rate and cell growth requiring increased synthesis of new proteins (doi.org/10.3390/cells8080869; doi: 10.1038/s41580-020-0272-6). Interestingly, the hypertrophy of the nucleolus is the histopathological diagnostic mark of the cancer transformation and tumor progression in clinical practice (doi: 10.3390/cells8010055).
The nuclear size is dependent on the balance of pressures and membrane tension. It has been speculated that nuclear size is maintained via balance of forces exerted on the outer and inner nuclear membranes, stronger pushing forces induce nuclear expansion in contrast to pressing forces encouraging nuclear shrinkage.

The manuscript should be edited.
